# Do Gender and Country of Residence Matter? A Mixed Methods Study on Lay Causal Beliefs about PTSD

**DOI:** 10.3390/ijerph191811594

**Published:** 2022-09-14

**Authors:** Caroline Meyer, Louisa Heinzl, Christina Kampisiou, Sofia Triliva, Christine Knaevelsrud, Nadine Stammel

**Affiliations:** 1Division of Clinical Psychological Intervention, Freie Universität Berlin, 14195 Berlin, Germany; 2Department of Psychology, University of Crete, 74100 Rethymno, Greece

**Keywords:** post-traumatic stress disorder, illness perceptions, explanatory models, trauma, cultural clinical research, mixed methods, intersectionality

## Abstract

Laypersons’ causal beliefs about mental disorders can differ considerably from medical or psychosocial clinicians’ models as they are shaped by social and cultural context and by personal experiences. This study aimed at identifying differences in causal beliefs about post-traumatic stress disorder (PTSD) by country and gender. A cross-sectional, vignette-based online survey was conducted with 737 participants from Germany, Greece, Ecuador, Mexico, and Russia. Participants were presented with a short unlabeled case vignette describing a person with symptoms of PTSD. Causal beliefs were assessed using an open-ended question asking for the three most likely causes. Answers were analyzed using thematic analysis. Afterwards, themes were transformed into categorical variables to analyze differences by country and by gender. Qualitative analyses revealed a wide range of different causal beliefs. Themes differed by gender, with women tending to mention more external causal beliefs. Themes also differed between the five countries but the differences between countries were more pronounced for women than for men. In conclusion, causal beliefs were multifaceted among laypersons and shared basic characteristics with empirically derived risk factors. The more pronounced differences for women suggest that potential gender effects should be considered in cross-cultural research.

## 1. Introduction

Posttraumatic stress disorder (PTSD) is a common mental disorder that can develop in the aftermath of experiencing a traumatic event such as assault, natural disasters, or accidents. PTSD comprises disrupting symptoms such as recurring memories of the traumatic event, avoidance of thoughts, negative changes in thinking and mood and hypervigilance [1]. Across the world, the lifetime prevalence of PTSD varies between 1% and 9%, with prevalence rates being influenced by political, historical, and cultural factors across countries. Especially in low-income and post-conflict countries, the prevalence of traumatic exposure and PTSD can be substantially higher [2,3,4]. While PTSD is caused by a preceding traumatic event, traumatic exposure does not necessarily lead to PTSD, indicating a multifactorial etiology [4]. Meta-analyses have identified several risk factors, differentiating between factors before, during, and after the trauma [5,6,7]. Among the most important risk factors were specific characteristics during the traumatic situation, such as perceived life-threat or dissociation and general life stress or lack of social support in the aftermath of experiencing trauma. The authors also identified several sociodemographic factors and experiences before the trauma, including being female, having a lower socio-economic status, belonging to a racial minority, prior trauma, childhood adversities, and history of psychopathology in the family.

Not unlike clinicians, laypersons hold beliefs about possible causes for their own symptoms or those of others. These lay causal beliefs show similarities to the multifactorial etiological models of clinicians and are often multilayered and complex. At the same time, they are less static and likely to change over time or in response to contextual factors and experiences [8,9]. As causal beliefs also shape beliefs about appropriate and effective treatments, they are highly relevant for clinical psychology [9,10]. Accordingly, the lack of understanding about the causes and correlates of mental illness can be a significant barrier to help-seeking [11,12].

Cultural differences have been the major research focus of causal beliefs in the past. So far, most studies on causal beliefs about PTSD have focused on refugee populations [13,14,15]. Qualitative studies have shown that refugees often hold various causal beliefs about mental distress. Commonly identified themes in these studies were traumatic life experiences, psychological causes, social causes, religious or spiritual causes, gender-related factors, and causes directly linked to being a refugee, such as flight experience, post-migration stressors, or discrimination [16,17,18,19,20,21,22,23,24]. In quantitative studies, traumatic experiences, interpersonal or social conflicts, being female, and aging were most commonly reported as risk factors [25,26,27,28].

Studies on causal beliefs about PTSD in Western laypersons are exceptionally scarce. Spoont, et al. [29] studied causal beliefs in predominantly male US veterans using the Illness Perception Questionnaire [30] and found higher agreement with psychosocial than with biological causal beliefs. In their study, agreement on biological causal beliefs was predicted by lower levels of education and higher age, but aside from biological and psychosocial factors, no other causal beliefs were assessed. Some quantitative studies comparing causal beliefs have also included both refugees and participants born in the respective Western host country. While all participants expressed causes related to the traumatic event and psychological causes, in some refugee populations, participants were more likely to express spiritual and biological causal beliefs compared to participants from their Western host country [19,27]. Similar results were found for patients from Iran compared to patients from Germany [31]. Cultural differences in causal beliefs have, for example, been attributed to specifics of mental health systems, to the role of religion in everyday life, or to refugee status [19,26,31]. Few studies have additionally considered gender-based differences, and overall results have been inconclusive. Qualitative studies targeting refugee populations have found differences in the way women and men perceive causes and possible treatments for mental illnesses [25,32]. While women tended to mention factors related to society, structural violence, gender-roles or lack of social support in the community, men tended to focus on intrapersonal characteristics, such as psychological or somatic causes. In contrast, a quantitative study with patients from Iran and Germany did not find any significant effect of gender on causal beliefs [31].

Clinical research in transcultural contexts presents unique challenges and therefore requires careful consideration of adequate methods and frameworks that can be used to answer complex research questions [33]. An interesting approach is presented by the intersectionality theoretical framework. According to this framework, multiple social categories (e.g., gender, ethnicity, race) intersect at the individual level of experience with large-scale social-cultural processes and institutionalized relationships resulting in power differentials [34,35]. In the context of mental health, this alludes to the intersection of multiple sources of influence on mental health and well-being [36]. By examining multiple aspects of identity simultaneously to study their impact on mental health, it offers a novel perspective for cultural clinical research [37,38]. Thereby, this approach can provide important insights into the health needs and experiences of marginalized populations and can contribute to improving health care practices [39]. Several studies have used the intersectional framework to study the combined effects of cultural factors and other aspects of social identity, such as age, class, sexual orientation, or gender [40,41,42,43,44,45]. 

The aims of this study were twofold. First, the study aimed at a broader understanding of lay causal beliefs in a cross-cultural sample from five countries. It was hypothesized that participants would support a wide range of causal beliefs about PTSD (hypothesis 1). Second, by using an intersectional approach, the study aimed at gaining a better understanding of how gender and country interact as correlates of causal beliefs. In line with previous studies, it was assumed that participants’ causal beliefs would differ by country of residence (hypothesis 2) and by gender (hypothesis 3). It was additionally assumed that there are interactions between the person’s gender and country of residence for causal beliefs about PTSD, thus that results would differ for male and female participants for each country (hypothesis 4).

## 2. Materials and Methods

A cross-sectional, online vignette-based study was conducted from February 2019 to May 2019. Laypersons in Germany, Russia, Ecuador, Mexico, and Greece were surveyed using Unipark [46]. The research ethics committee of Freie Universität Berlin approved this study (202/2018). 

### 2.1. Participants

Participants with a minimum age of 18 who lived in one of the five target countries at the time of the study could participate. The survey was accessible via a link that was distributed over internet platforms, social media advertisements and postings (e.g., Facebook, Twitter, VKontakte, Instagram), health-related online forums, institutional and university mailing lists, and local organizations. Recruitment strategies targeted participants of different ages and interest groups with diverse educational and occupational backgrounds to create a diverse sample. Before giving their informed consent, participants were informed about the aims of the study, its duration as well as privacy aspects. To increase the rate of consent and completion of the survey participants were offered a chance to participate in a raffle for shopping vouchers. The incentives were vouchers to widely used regional stores or companies that were attractive to a large sector of the population and could thus ensure a representative sample. Participants were informed beforehand that the survey was completely anonymous and apart from demographic data, no personal data would be collected. For entering the raffle, participants were directed to a second survey, where the email-address was collected independently from the main survey to ensure anonymity. For each country one voucher worth €100 and five vouchers worth €20 (or of equal worth in the corresponding national currency) were raffled among those who gave their email-address. 

### 2.2. Study Design

#### 2.2.1. Vignette

A short unlabeled case vignette describing a fictitious person with PTSD symptoms (according to the Diagnostic and Statistical Manual of Mental Disorders, Fifth Edition [DSM-5] [1]) was presented to the participants. To aid identification and to ensure that participants would answer as if they themselves were in a similar situation, the gender of the fictitious person in the vignette (Maria/Alexander) was matched to the gender of the participant:

“I would like you to imagine Maria, a fictitious person: Maria is about your age. For the past two months, she has been sleeping poorly. She often has vivid nightmares and therefore she frequently wakes up in the middle of the night feeling very frightened. She is jumpy and easily startled and has lost interest in activities she enjoyed before. This all began a few months ago, after an incident in which she saw her life in danger. Maria was out alone, when she was threatened with a knife and robbed by two armed men. After the incident, Maria felt numb for several days and then the nightmares began. She still sees the armed men clearly in her nightmares. When she hears an unexpected noise, she gets easily startled. Maria has been very afraid to go outside alone since the event and expects danger at all times. She tries not to think about the assault and does not want to talk to others about it.”

#### 2.2.2. Open-Ended Question

After answering two standardized questionnaires which listed various possible causes for Maria’s/Alexander’s symptoms [47,48], participants were asked to name the three causes they thought were the most likely for the problems described in the vignette: “Finally, please tell us which three factors you think are most responsible for Maria’s problems. You can do this by naming a possible cause from the previous list or by taking a cause that is not mentioned.”

#### 2.2.3. Quantitative Measures and Sociodemographic Variables

Participants gave information regarding their country of residence, age, years of education, and self-identified gender (“male”, “female”, or “diverse”). The vignette and all measures were administered in the respective national languages of the included countries (German, Greek, Spanish, and Russian). All material was subjected to a three-step approach to obtain valid translated measures. 1. Two bilingual mental health professionals translated and 2. back-translated the vignette and questionnaires for each language, and 3. agreed on a translation after discussing possible discrepancies. Afterwards, all involved translators and the authors (C.M., N.S., and C.K. [Christina Kampisiou]) discussed potentially ambiguous expressions to ensure that the interpretation would be as similar as possible for all participants. 

### 2.3. Data Analysis

Prior to all analyses, respondents who were assumed to have answered items randomly were excluded to ensure data quality. Following Meade and Craig [49], several indicators were determined for each participant to identify these so-called careless respondents by using the ‘careless’ package in R4.0.2 [50], including the duration of participation, LongString and the Psychometric Synonyms Index as overall measures as well as intraperson variance and Mahalanobis distance for each separate scale. Participants, who were flagged on one or more indicators, were rated independently by two of the authors (C.M., C.K. [Christina Kampisiou]). When careless data patterns became apparent, these persons were excluded from the study. In case of disagreement, a consensus was reached by discussion. For the remaining participants, the proportion of missing values was less than 0.5% for each variable. To avoid listwise deletion, missing values were dealt with using expectation-maximization-based single value imputation, as implemented in the Statistical Package for Social Sciences (SPSS), Version 27.0 [51]. Before conducting the thematic analysis, all answers given to the open-ended question were translated by bilingual clinical psychologists into the native language (German) of the two analyzing authors. 

#### 2.3.1. Qualitative Analysis

Thematic analysis is a relevant and flexible qualitative approach that is suitable in research conducted by teams and in the analyses of large qualitative data sets [52]. In this study, two researchers (C.M. and L.H.) analyzed the data in a rigorous and methodical manner. The coding was conducted inductively as there were no presumptions regarding the themes that might emerge from the analysis. As the format of the question encouraged participants to give short answers with a clear focus and did not give a lot of contextual information, coding was conducted on a semantic level. Both C.M and L.H. are clinical psychologists and have previously worked with patients with trauma-related disorders from several cultural backgrounds. 

Following recommendations by Braun and Clarke [52], at first, the researchers read through the participants’ answers several times to familiarize themselves with the data. Afterwards, initial codes were generated independently by the two researchers to aggregate similar statements. After 25% of the data had been coded, codes were discussed between the researchers and a preliminary code system was developed. Based on the preliminary code system the entire qualitative data set was coded independently by both researchers, who continuously added new codes which they noticed in the data. Subsequently, the coded segments were compared, and discrepancies were discussed and agreed upon. L.H. and C.M. then jointly searched for themes into which the codes could be organized (first-order themes), which were again grouped into higher-order themes (second-order themes). After the initial creation of themes, they were reviewed again and the resulting thematic map was presented to two co-authors (N.S., C.K. [Christina Kampisiou]) to discuss controversies and verify the coherence of the themes. Finally, all themes were defined and named and a hierarchical system of first- and second-order themes resulted. The qualitative analysis was performed using MAXQDA 2020 [53].

#### 2.3.2. Quantitative Analysis

All first- and second-order themes resulting from the thematic analysis were transformed into dichotomous variables (0 = theme not mentioned, 1 = theme mentioned) to allow for quantitative data analysis. Frequencies were calculated for all first- and second-order themes. Afterwards, the second-order themes were analyzed for differences concerning country of residence and gender using Fisher’s exact test. To determine whether a person’s gender interacted with the effect of the country of residence, Fisher’s exact test was performed separately for men and women. Cohen’s ω was calculated to measure the size of the effect, with ω = 0.1 indicating a small effect, ω = 0.3 indicating a medium effect, and ω = 0.5 indicating a large effect [54]. Variables with assigned *p*-values < 0.05 were considered statistically significant. For post-hoc testing, Hochberg’s pairwise testing was used to correct for multiple testing as suggested in non-balanced designs [55,56]. All statistical analyses were performed with R4.0.2 [50]. Two participants self-identified their gender as “diverse” and were excluded from all analyses including gender as a variable due to the small group size.

## 3. Results

### 3.1. Sample Characteristics

The vignette was shown to N = 853 participants. Of those, 97 did not complete the survey and another 19 participants were excluded for giving careless responses. The interrater reliability on exclusion due to careless response patterns was excellent ([κ = 0.84] [57]). The final sample consisted of N = 737 participants (492 females, 66.8%). The age ranged from 18 to 78 years (M = 36.2, SD = 13.9). Sample characteristics for the total sample as well as by country are given in Table 1. The sample has been described in more detail elsewhere [58].

### 3.2. Qualitative Data Analysis

In total, 2404 statements about causal beliefs were coded and assigned to the category system by thematic analysis. Forty-nine first-order themes emerged from the initial codes and were categorized into eight second-order themes that are described in more detail in the following part. A comprehensive list of all first- and second-order themes can be found in Table 2. 

**Emotional, cognitive, and behavioral reactions**: Participants mentioned a wide range of emotional or behavioral reactions, such as feelings of anxiety and humiliation, a reduced sense of security, negative thoughts, or problems with sleeping (“Maria’s mental health”, “difficulties, finding trust in other people”, “feelings of shame and guilt”, “she experienced herself as powerless”).

**Characteristics of the person**: Participants made references to certain characteristics of the person described in the vignette in several ways. Several participants mentioned the person’s character in general or certain personality traits as likely causes (“anxious personality”, “her personality”, “he is a weak character”, “shyness”, “she is generally very sensitive”) along with several biological factors lying within the person (“illness”, “problems with immunity”, “genetic disposition”, “the process of aging”), and their attitude (“nihilistic thinking”, “negativism”, “her attitude towards life”). Some participants also mentioned previous experiences of the person that may have contributed to the problems (“problems in childhood”, “negative experiences in the past”, “she might have been abused as a child, and the attack brought those memories to the surface”). One person also mentioned education as a potential cause (“level of education”).

**Reference to the event**: Participants made references to the attack described in the vignette in several ways. Some merely referred to the event (“the attack”, “being assaulted”, “a sudden and unexpected attack”, “the event that threatened his life”, “it was the men with the knives”), while others focused on the violent nature of the attack (“being the victim of aggression”, “experiencing violence”) or clarified that they considered the event to be traumatic (“traumatic event”, “he had a traumatizing experience”). Some participants also commented on the behavior in the situation (“she was unable to react”, “defensive behavior encourages an attack”). 

**Social and societal factors**: Participants mentioned several issues that lay outside the person and can be found either on the societal level (“poverty”, “increased crime in the area”, “she has done something that is considered taboo for the society in which she lives”), or in the direct surroundings of the person (“living in an environment where she feels insecure”, “living conditions in the neighborhood”, “fear of being insulted for being involved in the attack”, “social surroundings”). Several participants also mentioned lack of support, either on a professional level (“lack of psychological support services in his local community”, “poor medical care”, “lack of psychotherapy”) or on an informal level (“lack of social contacts, too little help from others”, “lack of stable, close relationships and family ties that help to ease anxiety”, “she did not get any help after the event”). 

**Inappropriately dealing with distress**: Several participants mentioned factors related to the person’s dealing with distress, mostly commenting on inappropriate or unfavorable dealing with the situation. This included a general lack of self-help strategies (“not providing reassurance, e.g., through mindfulness techniques, relaxation”, “he cannot handle emotions”), but also not seeking help proactively (“he refuses to take help from others”, “maybe she does not know where to get help”, “lack of courage to seek professional help”) and the lack of communicating about their problems with others (“she doesn’t want to talk to anyone about the problem, which makes them seem bigger and scarier, thereby making her worse”, “not having talked to anyone about it”) most likely lead to the person’s problems. Participants also mentioned several specific strategies they thought were inappropriate, such as substance abuse (“consumption of alcohol”, “drug abuse”, “addiction”), a lack of processing or avoidance (”he is afraid of facing his problems”, “his psychological trauma has not been processed”, “she refuses to accept and talk about what happened”), and not being able to overcome the past (“having ongoing thoughts/worries about this event”, “she cannot forget what happened to her”). 

**Problems in everyday life**: Participants mentioned several problems in everyday life that could have contributed to the problems, for example, work-related problems (“problems at the workplace”, “being overworked”, “unemployment”) or problems with their family (“problems in the family”, “worries about his family”, “conflicts in the family”). Some participants also commented on this on a broader level, such as being in a difficult situation (“Maria was already in a difficult place”, “he has a boring life”, “having an unsuccessful life”) or pursuing a problematic lifestyle (“constant stress”, “problematic habits”, “maybe she does not eat healthy”). 

**Mental disorders**: In addition to nonspecific mentions of emotional and behavioral reactions, participants also made reference to several mental disorders, such as anxiety disorders (“panic attacks”, “phobias”), depression (“depression”), or trauma-related disorders (“psychological trauma”, “she has post-traumatic stress”, “PTSD”).

**Spirituality**: Few participants mentioned explicitly spiritual or religious issues as the most likely causes (“karma”, “crisis of faith”, “religion”), but several participants mentioned fate as a possible cause (“bad luck”, “fate”). 

In summary, a wide range of causes was found in the answers, indicating that participants held a variety of different explanations for the problems depicted in the vignette. These themes included also other domains of explanatory models, such as labels, specific symptoms, and help-seeking recommendations. Also, most participants expressed support for more than one of the second-order themes, indicating that many held multicausal beliefs. For a more in-depth understanding, this was further explored in the quantitative analysis. 

### 3.3. Quantitative Data Analysis

All first- and second-order themes resulting from the thematic analysis were transformed into dichotomous variables for quantitative data analysis. On average, participants mentioned 2.4 second-order themes (min = 0, max = 6). A large majority of participants mentioned at least two second-order themes (89%) and almost half (43.8%) mentioned even three or more second-order themes. Table 2 reports the percentage of participants mentioning each theme for all first- and second-order themes. Among the second-order themes, *emotional, cognitive, and behavioral reactions* was most frequently mentioned (61.3%), followed by *characteristics of the person* (46.1%), and *reference to the event* (39.3%). Less frequently named were the themes *social and societal factors* (28.5%), *inappropriately dealing with distress* (24.8%), *problems in everyday life* (16.7%), and *mental disorders* (16.6%). Least frequently mentioned was the theme *spirituality* (5.0%). Themes were relatively independent from each other, only weakly correlated (r = −0.26 to r = 0.07), and showed no systematic patterns in overlap (Appendix A). 

#### 3.3.1. Themes by Country of Residence

To investigate differences in causal beliefs across the five countries, the frequencies of all second-order themes were compared using Fisher’s exact test (see Table 3 for details). For six out of eight second-order themes, significant differences were found between the countries. However, post-hoc tests revealed relatively few significant differences when testing pairwise. Although causal beliefs differed between the five countries, in most cases, the differences were only significant for the countries on each extreme of the continuum (e.g., for the second-order theme *characteristics of the person* when comparing Mexico with the lowest rate of mentions [30.0%] vs. Greece with the highest rates of mentions [55.2%]; see Table 3). In summary, no systematic pattern emerged and the overlap between the five countries was considerable. 

#### 3.3.2. Themes by Gender

To investigate differences in causal beliefs among women and men, the frequencies of all second-order themes were compared using Fisher’s exact test (see Table 4 for details). For three out of eight second-order themes, significant differences were found between women and men. As in the overall sample, the second-order theme, *emotional, cognitive, and behavioral reactions* was most frequently mentioned for both genders, followed by *characteristics of the person,* and *reference to the event*. In direct comparison, however, a significantly higher percentage of women mentioned the themes *r**eference to the event, social*
*and societal factors*, and *i**nappropriately dealing with distress* compared to men.

#### 3.3.3. Comparison by Country and Gender

To investigate possible interactional effects of gender and country of residence, Fisher’s exact test to test for differences between the five countries was performed separately for men and women and Cohen’s ω was calculated to measure the size of the effect. 

Results on the country-level differed between the male and the female sample, indicating a possible moderating effect of gender on the differences between countries (Table 5). For three second-order themes, there were significant differences between the countries in the female sample but not in the male sample, namely *reference to the event* (*p* < 0.001, ω = 0.23)*, social and societal factors* (*p* = 0.004, ω = 0.17)*,* and *problems in everyday life* (*p* = 0.002, ω = 0.19). The second-order theme *inappropriately dealing with distress* was the only one that showed significant differences between the countries for both samples with equally high effect sizes (*p* < 0.001, ω = 0.24 for females vs. *p* < 0.001, ω = 0.25 for males). Besides, significant differences between the countries were not found in the male sample for any of the second-order themes.

## 4. Discussion

### 4.1. Causal Beliefs in Laypersons

This vignette study explored causal beliefs in laypersons for PTSD using a mixed methods approach combining qualitative and quantitative analyses. The first aim of this study was to gain a better understanding of lay causal beliefs in a non-refugee, cross-cultural sample. The qualitative thematic analysis resulted in 49 first-order themes that were categorized into eight second-order themes, namely *emotional, cognitive, and behavioral reactions; characteristics of the person; reference to the event; social and societal factors; inappropriately dealing with distress; problems in everyday life; mental disorders;* and *spirituality.*

The first major finding of this study is that participants held a wide variety of causal beliefs, with most participants focusing on psychological causes, on the traumatic experience, and on social causes, often expressing beliefs from more than one domain at the same time. So far, studies on lay causal beliefs about PTSD have mainly targeted refugee populations from various areas across the world. Aside from the aspects that were specifically linked to flight or migration, the themes found in this study were very similar to the themes found in refugee samples [16,17,18,19,20,21,22,23,24]. The analysis of frequency and coexistence of second-order themes revealed that most participants in this study also mentioned two or more second-order themes when asked for the most likely causes of the problems described in the vignette. This further supports the assumption of coexisting instead of competing causal beliefs [19,59], indicating that laypersons hold multifactorial models similar to clinicians’ models. However, while overlap was considerate, the themes showed no systematic patterns. This indicates that not only causal beliefs from similar domains (e.g., psychosocial causes) overlap, but that participants shared causal beliefs from several different domains. 

Overall, most of the causes mentioned by the participants resembled causes and risk factors for PTSD that have been identified by meta-analyses, e.g., certain characteristics of the traumatic event, general life stress, lack of social support, lower socio-economic status, prior trauma, childhood adversities, and being female [5,6,7]. The results thus indicate that clinicians’ etiological models for PTSD might be plausible for, and shared by, laypersons. However, regarding the congruence of lay and professional models, it was also noticeable that although participants were explicitly asked about causes, several participants mentioned aspects that might as well be classified as the problem itself and referred to nonspecific symptoms or mental illnesses. The distinction between cause and effect may not be as clear for laypersons as it is stated in professional models. This result is consistent with findings suggesting that several domains of explanatory models might be interwoven [60]. Also, while the potentially traumatic event is central in etiological models of PTSD and accordingly was explicitly described in the vignette as the starting point for the problems, less than half of the participants made a reference to the event described in the vignette. At the same time, many participants mentioned aspects that were not mentioned in the vignette, such as overworking or problems in the family. Seemingly, besides recognizing the importance of a potentially traumatic event [27,61], many laypersons consider other factors to be more important for developing PTSD, even making inferences beyond the information given in the vignette.

### 4.2. Country and Gender in Causal Beliefs

The second aim of this study was to analyze differences in the frequency of the expressed second-order themes by country of residence and by gender, and possible interactions between both factors. It was assumed that participants’ causal beliefs would differ in frequency by country of residence because of differing cultural, historical, and political settings and by gender. Following an intersectional approach, it was further assumed that differences between countries would be different for male and female participants.

Our results indicate that the majority of mentioned causal beliefs differed between the five countries, but only a few differences were significant when compared pairwise. This is in line with previous studies that have found differences between ethnic groups for specific characteristics, such as higher expressions of spiritual causal beliefs, but also stress the considerable heterogeneity within groups [19,62]. It is possible that, in contrast to other studies that have cross-culturally compared causal beliefs, the countries chosen for this study were more similar in terms of cultural and contextual aspects. Overall, the results support the idea that various models can be found across different populations and that nationality or ethnicity predict causal beliefs inaccurately [19,27,63]

Differences between causal beliefs were more pronounced between male and female participants. Noticeably, two out of the three themes that were more frequently mentioned by women were external factors (*reference to the event* and *social and societal factors*). So far, to our knowledge, there are no studies analyzing gender differences in causal beliefs in laypersons from non-refugee populations. In other populations, the results for gender effects in causal beliefs are inconsistent [19,31,32]. Our results are in line with results from qualitative studies targeting refugee populations that have suggested that women were more likely to mention factors related to society, structural violence, gender-roles, or lack of social support in the community, while men were more likely to mention intrapersonal characteristics, such as psychological or somatic causes. However, no differences between women and men were found in a sample surveying patients from Iran and Germany when asked for causes of their own symptoms [31]. It is possible that the effects are at least partly due to the chosen methodology using a case vignette and, thereby, are focused mainly on the perception of others which might enhance attributional bias. For causal attribution of rape, for example, a gender effect is well established in the literature. Attributional studies have found that women are more likely to express external causes for rape, such as circumstances or the perpetrator, while men were more likely to attribute the event internally, that is, they blame the victim [64,65]. Our results suggest that the findings on gender effects in rape scenarios may be extended also to PTSD symptoms following interpersonal trauma. Women in our study tended to be less focused on internal or psychological causes than men. 

Finally, this study aimed at illustrating intersectional effects in cross-cultural research focusing on the interactions of gender and country of residence. The results of this study indicate that gender matters in the comparison of countries, thereby supporting the intersectional approach. For several factors, differences at the national level were more pronounced for women compared to men. These differences were found especially for the second order themes related to the event, to social and societal factors, and to problems in everyday life. One possible explanation for this result is that the differences between the countries are more relevant for women and less so for men. One important factor might be, for example, how commonly women experience violence. This can differ considerably between countries and can depend on, for example, gender roles and country specific policies [66,67] which may lead to more accepting attitudes towards violence in women [68,69]. This might, in turn, affect causal beliefs and explain why differences for women were more pronounced than for men in our sample. This is an interesting finding but the results must be interpreted with care due to the small effects and the qualitative methodology. It must also be noted that the sample size differed between genders and that there is more data on women. However, it is possible that by choosing a mixed methods approach, gender effects that are overlooked or not reported in quantitative designs were made visible. In this study, the most important differences between males and females ran along the lines of the internal and external attribution of symptoms. This aspect and possible implications for related concepts, such as stigma and help-seeking intentions, should be considered in future studies. 

In summary, the second major finding of this study is that there were noteworthy gender effects in this study. There were significant differences in causal beliefs between male and female participants. Also, differences between countries were more pronounced in women and much smaller or non-existent among men. This highlights the importance of including gender-aspects in transcultural research and underlines the problems that arise from single-gender samples or samples with highly unbalanced gender distributions that may fail to cover gender-specific experiences. Intersectional approaches pose an interesting approach for research in transcultural settings as they allow researchers to equally consider cultural and gender aspects. 

### 4.3. Limitations

Several limitations need to be considered. First, data collection was carried out via convenience sampling through an online survey and included only a small selection of countries. As the sample was not balanced concerning gender and country and participants were relatively young and well educated, one should be careful to generalize the results to other socioeconomic groups or countries. Although the sample is not representative, it nevertheless reflects typical sample characteristics that have been found in online-based surveys [70,71,72]. 

Second, by the nature of the chosen method, all results are highly dependent on the case vignette. It must be noted that, although, vignette studies are a well-established method for assessing mental health beliefs [73], they cannot simulate symptoms of PTSD. Furthermore, as a robbery was chosen as the traumatic incident, results might differ for cases with other traumatic experiences, such as sexual assault or combat [74] and all effects must be interpreted as beliefs that participants hold about their own gender, as female participants were presented with a female case vignette and male participants were presented with a male case vignette to ensure that participants would identify as much as possible with the case. In addition, the participants were presented with a structured questionnaire before answering the open-ended question, which may have influenced their answers. 

Third, the transcultural approach posed several challenges. Prior to the thematic analysis, all statements were translated into the first language of the analyzing authors to ensure coherent coding. While thematic analysis focuses on the content of the statements rather than on linguistic details, it is still possible that by using the translated statements, nuances in the statements were missed that could have led to different coding. Also, both raters have been born and raised in one of the five countries and may therefore be biased in their coding and ratings as qualitative research methods always highly depend on the perception of the raters. 

Finally, for an in-depth understanding of possible interactions between country and gender, we chose a narrow focus for our analyses and the chosen method did not allow the inclusion of further cultural or sociodemographic covariates. In addition, all quantitative analyses should be considered exploratory in nature and additional studies are needed to confirm the results.

## 5. Conclusions

Largescale mixed methods studies can be a valuable approach for capturing health beliefs in the widest possible way to reduce the problem that cultural presumptions influence the results while at the same time it enables researchers to draw conclusions about frequencies and potential correlates. The results of this study indicate that lay persons hold a wide variety of possible beliefs about PTSD that largely overlap with risk factors identified by recent clinical research. While there were only a few significant differences between countries when compared pairwise, beliefs clearly differed between women and men and differences between countries were also more pronounced for women compared to men. Using an intersectional approach in cross-cultural research can lead to a better understanding of the interrelationships between several factors influencing cultural differences. The results further suggest that gender should be considered when researching causal beliefs in transcultural contexts. Moreover, practitioners should be aware and mindful of potential gender differences in causal beliefs and consider these and their potential interactions with cultural factors in psychotherapy and counseling, especially when culturally adapting interventions. Future research should further expand to other social characteristics and include factors potentially leading to disadvantages in mental health care, such as race, class, age, disability, or sexual identity. With concern to the intersectional approach, these factors should not be studied singularly, but also the intersections of these factors should be considered where possible. 

## Figures and Tables

**Table 1 ijerph-19-11594-t001:** Sample description by current country of residence.

	Total (N = 737)	Germany (N = 261)	Russia (N = 134)	Ecuador (N = 99)	Mexico (N = 60)	Greece (N = 183)
**Gender**						
Female	492 (66.8%)	184 (70.5%)	100 (74.6%)	54 (54.5%)	37 (61.7%)	117 (63.9%)
Male	243 (32.9%)	76 (29.1%)	34 (25.4%)	45 (45.5%)	23 (38.3%)	65 (35.5%)
Diverse	2 (0.3%)	1 (0.4%)	0 (0.0%)	0 (0.0%)	0 (0.0%)	1 (0.6%)
**Age** (mean [SD])	36.2 (13.9)	36.2 (15.1)	36.8 (14.2)	35.0 (12.2)	31.3 (10.7)	37.9 (13.4)
**Years of education** (mean [SD])	15.8 (4.7)	15.7 (5.0)	16.4 (3.1)	15.4 (5.5)	13.5 (7.1)	16.4 (3.6)

**Table 2 ijerph-19-11594-t002:** Percentage of participants mentioning each causal belief according to the thematic analysis of responses to naming the most likely causes.

Second-Order Theme	First-Order Theme	N	%
**Emotional, cognitive, and behavioral reactions**		**452**	**61.3**
	Stress or worries	267	36.2
	Anxiety	162	22.0
	Mental burden	143	19.4
	Reduced sense of security	92	12.5
	Worldview shattered	26	3.5
	Feeling helpless/vulnerable	23	3.1
	Feelings of humiliation	12	1.6
	Feeling nervous/tense	12	1.6
	Problems with sleeping	11	1.5
	Negative thoughts	10	1.4
	Reexperience	3	0.4
	Subconscious	2	0.3
	Pain	1	0.1
**Characteristics of the person**		**340**	**46.1**
	Personality	133	18.0
	Biological factors	82	11.1
	Previous experiences	77	10.4
	Attitude	72	9.8
	Low self-esteem	18	2.4
	Resilience	11	1.5
	Weakness	10	1.4
	Predisposition	9	1.2
	Education	1	0.1
**Reference to the event**		**290**	**39.3**
	Traumatic event	150	20.4
	Attack/assault	122	16.6
	Experiencing violence	31	4.2
	Stress related to the attack	10	1.4
	Behavior in the situation	8	1.1
**Social and societal factors**		**210**	**28.5**
	Lack of support	105	14.2
	Lack of therapy/professional support	52	7.1
	Societal problems	47	6.4
	Financial problems	23	3.1
	Social environment	21	2.8
	No legal consequences	1	0.1
**Inappropriately dealing with distress**		**183**	**24.8**
	Lack of communication about the attack	48	6.5
	Lack of processing the attack	42	5.7
	Lack of self-help strategies	36	4.9
	Not seeking help proactively	30	4.1
	Own behavior	25	3.4
	Avoidance	24	3.3
	Substance abuse	13	1.8
	Not being able to overcome the past	8	1.1
	Lack of positive experiences	1	0.1
**Problems in everyday life**		**123**	**16.7**
	Problems in everyday life	123	16.7
**Mental disorders**		**122**	**16.6**
	Shock	62	8.4
	Trauma-related disorders	40	5.4
	Anxiety disorders	**14**	**1.9**
	Depression	7	0.9
	Mental disorder	6	0.8
**Spirituality**		**37**	**5.0**
	Spirituality	37	5.0

**Table 3 ijerph-19-11594-t003:** Percentage of participants mentioning the second-order theme according to the thematic analysis presented by current country of residence. *p*-values were calculated with Fisher’s exact test.

	Germany (N = 261)	Russia (N = 134)	Ecuador (N = 99)	Mexico (N = 60)	Greece (N = 183)	*p*-Value	Sig. Pairwise Comparisons ^1^
Emotional, cognitive, and behavioral reactions	151 (57.9%)	88 (65.7%)	63 (63.6%)	32 (53.3%)	118 (64.5%)	0.295	
Characteristics of the person	117 (44.8%)	53 (39.6%)	51 (51.5%)	18 (30.0%)	101 (55.2%)	0.004 **	Greece > Mexico **
Reference to the event	132 (50.6%)	40 (29.9%)	41 (41.4%)	28 (46.7%)	49 (26.8%)	<0.001 ***	Mexico > Greece * Germany > Greece *** Germany > Russia ***
Social and societal factors	68 (26.1%)	56 (41.8%)	20 (20.2%)	16 (26.7%)	50 (27.3%)	0.003 **	Russia > Germany * Russia > Ecuador **
Inappropriately dealing with distress	96 (36.8%)	35 (26.1%)	16 (16.2%)	15 (25.0%)	21 (11.5%)	<0.001 ***	Germany > Ecuador ** Germany > Greece *** Russia > Greece **
Problems in everyday life	30 (11.5%)	24 (17.9%)	14 (14.1%)	8 (13.3%)	47 (25.7%)	<0.001 ***	Greece > Germany **
Mental disorders	38 (14.6%)	31 (23.1%)	21 (21.2%)	12 (20.0%)	20 (10.9%)	0.024 *	
Spirituality	14 (5.4%)	9 (6.7%)	1 (1.0%)	2 (3.3%)	11 (6.0%)	0.252	

Note. ^1^ Hochberg’s pairwise testing was performed to correct for multiple testing. “>” indicates that participants from this country were more likely to mention this second-order theme when compared pairwise.* *p* < 0.05, ** *p* < 0.01, *** *p* < 0.001.

**Table 4 ijerph-19-11594-t004:** Percentage of participants mentioning the second-order theme according to the thematic analysis presented by gender. *p*-values for gender differences were calculated with Fisher’s exact test.

	Female (N = 492)	Male (N = 243)	*p*-Value
Emotional, cognitive, and behavioral reactions	305 (62.0%)	145 (59.7%)	0.574
Characteristics of the person	219 (44.5%)	120 (49.4%)	0.238
Reference to the event	209 (42.5%)	80 (32.9%)	0.013 *
Social and societal factors	155 (31.5%)	54 (22.2%)	0.009 **
Inappropriately dealing with distress	134 (27.2%)	49 (20.2%)	0.037 *
Problems in everyday life	77 (15.7%)	46 (18.9%)	0.294
Mental disorders	83 (16.9%)	39 (16.0%)	0.833
Spirituality	20 (4.1%)	17 (7.0%)	0.106

Note. N = 735. Two participants self-identified as “diverse” and were excluded from the analysis due to the small group size. * *p* < 0.05, ** *p* < 0.01.

**Table 5 ijerph-19-11594-t005:** Percentage of participants mentioning the second-order theme according to the thematic analysis presented by current country of residence for female (upper table) and male (lower table) participants. *p*-values were calculated with Fisher’s exact test.

FEMALES							
	Germany (N = 184)	Russia (N = 100)	Ecuador (N = 54)	Mexico (N = 37)	Greece (N = 117)	*p*-Value	Cohen’s ω
Emotional, cognitive, and behavioral reactions	107 (58.2%)	67 (67.0%)	35 (64.8%)	21 (56.8%)	75 (64.1%)	0.560	0.08
Characteristics of the person	83 (45.1%)	39 (39.0%)	24 (44.4%)	11 (29.7%)	62 (53.0%)	0.098	0.13
Reference to the event	96 (52.2%)	31 (31.0%)	29 (53.7%)	20 (54.1%)	33 (28.2%)	<0.001 ***	0.23
Social and societal factors	46 (25.0%)	46 (46.0%)	13 (24.1%)	11 (29.7%)	39 (33.3%)	0.008 **	0.17
Inappropriately dealing with distress	73 (39.7%)	27 (27.0%)	9 (16.7%)	8 (21.6%)	17 (14.5%)	<0.001 ***	0.24
Problems in everyday life	20 (10.9%)	16 (16.0%)	6 (11.1%)	3 (8.1%)	32 (27.4%)	0.003 **	0.19
Mental disorders	29 (15.8%)	24 (24.0%)	11 (20.4%)	6 (16.2%)	13 (11.1%)	0.126	0.12
Spirituality	6 (3.3%)	7 (7.0%)	0 (0.0%)	1 (2.7%)	6 (5.1%)	0.260	0.10
**MALES**							
	**Germany** **(N = 76)**	**Russia** **(N = 34)**	**Ecuador** **(N = 45)**	**Mexico** **(N = 23)**	**Greece** **(N = 65)**	** *p* ** **-Value**	**Cohen’s ω**
Emotional, cognitive, and behavioral reactions	43 (56.6%)	21 (61.8%)	28 (62.2%)	11 (47.8%)	42 (64.6%)	0.651	0.10
Characteristics of the person	34 (44.7%)	14 (41.2%)	27 (60.0%)	7 (30.4%)	38 (58.5%)	0.061	0.19
Reference to the event	35 (46.1%)	9 (26.5%)	12 (26.7%)	8 (34.8%)	16 (24.6%)	0.057	0.20
Social and societal factors	21 (27.6%)	10 (29.4%)	7 (15.6%)	5 (21.7%)	11 (16.9%)	0.342	0.14
Inappropriately dealing with distress	23 (30.3%)	8 (23.5%)	7 (15.6%)	7 (30.4%)	4 (6.2%)	0.002 **	0.25
Problems in everyday life	10 (13.2%)	8 (23.5%)	8 (17.8%)	5 (21.7%)	15 (23.1%)	0.515	0.11
Mental disorders	9 (11.8%)	7 (20.6%)	10 (22.2%)	6 (26.1%)	7 (10.8%)	0.190	0.16
Spirituality	8 (10.5%)	2 (5.9%)	1 (2.2%)	1 (4.3%)	5 (7.7%)	0.550	0.12

Note. N = 735. Two participants self-identified as “diverse” and were excluded from the analysis due to the small group size. ** *p* < 0.01, *** *p* < 0.001.

## Data Availability

The data are not publicly available due to their containing information that could compromise the privacy of research participants. The statistical code and results are available at the following link on Open Science Framework (OSF): https://osf.io/te3d2/ (accessed on 7 September 2022).

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
