# Peer review of "Do Gender and Country of Residence Matter? A Mixed Methods Study on Lay Causal Beliefs about PTSD"

_ijerph, 2022, doi:10.3390/ijerph191811594_

Round 1
Reviewer 1 Report
This is an interesting and well-written article. I also impressed at the detailed and exquisite study design. The major concern is its scientific aspect of this study.
1. This study used a picture and a scenario to simulate a condition of PTSD. It is doubtful that the effect of simulation could correlate the real situation of mental trauma.
2. The result of the study is predictable. Laypersons come from different countries with different characteristics and backgrounds will show various responses to a situation. It is hard to believe these subjects would present a similar response.
3. Regarding the baseline characteristics of laypersons, using only country and gender as the variables is overly simplified. I believe there are more personal variables that may alter the result of the study.
Author Response
Reviewer 1: Comments and Suggestions for Authors
This is an interesting and well-written article. I also impressed at the detailed and exquisite study design. The major concern is its scientific aspect of this study.
Author response: Thank you for the positive evaluation of our article. We have addressed each of your comments below.
- This study used a picture and a scenario to simulate a condition of PTSD. It is doubtful that the effect of simulation could correlate the real situation of mental trauma.
Author response: Thank you for this comment. We agree that a vignette study cannot simulate a trauma condition or symptoms of trauma. However, this was not the aim of our study. We aimed at a better understanding of causal beliefs about PTSD in laypersons and have decided to use this methodological approach as vignette studies are a well-established method for assessing mental health beliefs in laypersons (Wei et al., 2015). We have added this information in the limitations section.
Changes in manuscript:
Limitations: (l. 523): “It must be noted that, although, vignette studies are a well-established method for assessing mental health beliefs (Wei et al., 2015), they cannot simulate symptoms of PTSD.”
- The result of the study is predictable. Laypersons come from different countries with different characteristics and backgrounds will show various responses to a situation. It is hard to believe these subjects would present a similar response.
Author response: Thank you for your comment. We do not agree that the results of our study were predictable. So far, only few studies have targeted causal beliefs about PTSD in non-refugee laypersons, therefore our results have not yet been the subject of research, and little is known about cross cultural differences within Western countries. In fact, one of the striking results of our study is that for male respondents, the differences between countries were less pronounced or even non-significant. We, therefore, argue that our manuscript does make an important contribution to the field of cultural clinical research as it emphasizes the need to include gender in the research design and analysis.
- Regarding the baseline characteristics of laypersons, using only country and gender as the variables is overly simplified. I believe there are more personal variables that may alter the result of the study.
Author response: Thank you for your comment. We agree that it would be interesting to include more than two variables and have already mentioned this aspect in the conclusion (l. 561). However, in this manuscript, we chose a narrow focus using a mixed methods approach to allow for a more in depth understanding of the specific interactions of country of residence and gender. The methods used, therefore, did not allow for inclusion of more variables. To further emphasize this important aspect, we have added this also in the limitations section.
For more information on the effect of other personal variables, you may refer to another publication by our working group.
Meyer, C.; Kampisiou, C.; Triliva, S.; Knaevelsrud, C.; Stammel, N., Lay causal beliefs about ptsd and cultural correlates in five countries. Eur J Psychotraumato 2022, 13, (1), 2029333, https://doi.org/10.1080/20008198.2022.2029333.
Changes in manuscript:
Limitations: (l. 541): “Finally, for an in-depth understanding of possible interactions between country and gender, we chose a narrow focus for our analyses and the chosen method did not allow to include further cultural or sociodemographic covariates.”
Reviewer 2 Report
Very interesting topic, well presented with an adequate methodology.
Just to suggest in the results (3.2.1 Thematic analysis: Causal beliefs about PTSD), present the categories identified in a diagram or table.
Author Response
Reviewer 2: Comments and Suggestions for Authors
Very interesting topic, well presented with an adequate methodology.
Just to suggest in the results (3.2.1 Thematic analysis: Causal beliefs about PTSD), present the categories identified in a diagram or table.
Author response: Thank you very much for your positive evaluation of out manuscript and your suggestion. We have already included a complete overview over all themes in the section presenting the quantitative results (see Table 2). We have referred to this in the text as follows “A comprehensive list of all first- and second-order themes can be found in Table 2” (l. 250). As all themes are listed in table 2, we decided to avoid redundant information in the manuscript and refrained from adding a diagram.
Reviewer 3 Report
In this study, Meyer et al aimed at a broader understanding of lay causal beliefs in a cross-cultural sample from five countries and gaining a better understanding of how gender and country interact as correlates of causal beliefs. The subject of the work is relevant to the special issue: Defining, Assessing, and Treating Stress-Related Disorders across Cultures. The manuscript is of good impact, well conceptualized, and clearly written.
Moreover:
- In abstract the authors should describe the aim of the study.
- In Materials and Methods section the authors should describe which social networking sites and forums they used, add a description of the recruitment process. "Participants could enter a prize drawing at the end of the survey" - How took a place a prize drawing and what were the prizes?
"Prior to all analyses, respondents who were assumed to have answered items randomly were excluded to ensure data quality." - How was it verified that the respondents answered randomly?
Lines 176: The authors should describe the abbrevation of SPSS 27.
In Results section the authors should not start sentences with "N = ...". The authors should describe the tables.
- Some conclusions should be transferred to limitations and implications to practice.
Author Response
Reviewer 3: Comments and Suggestions for Authors
In this study, Meyer et al aimed at a broader understanding of lay causal beliefs in a cross-cultural sample from five countries and gaining a better understanding of how gender and country interact as correlates of causal beliefs. The subject of the work is relevant to the special issue: Defining, Assessing, and Treating Stress-Related Disorders across Cultures. The manuscript is of good impact, well conceptualized, and clearly written.
Author response: First, we would like to thank you for your positive evaluation of the manuscript and for the attentive review of our manuscript and your helpful comments.
- In abstract the authors should describe the aim of the study.
Author response: Thank you for your comment. We have added a line describing the aim of the study in the abstract.
Changes in manuscript:
Abstract: (l. 10): “This study aimed at identifying differences in causal beliefs about post-traumatic stress disorder (PTSD) by country and gender.”
- In Materials and Methods section the authors should describe which social networking sites and forums they used, add a description of the recruitment process. "Participants could enter a prize drawing at the end of the survey" - How took a place a prize drawing and what were the prizes?
Author response: Thank you for your comment. We have added a more detailed description in the Methods section.
Changes in manuscript:
Methods: (l. 115): “The survey was accessible via a link that was distributed over internet platforms, social media advertisements and postings (e.g. Facebook, Twitter, VKontakte, Instagram), health-related online forums, institutional and university mailing lists, and local organizations. Recruitment strategies targeted participants of different ages and interest groups with diverse educational and occupational backgrounds to create a diverse sample.”
Methods: (l. 121): “As a motivation to participate in the study, and to complete the survey, shopping vouchers were raffled among participants. For prizes we chose vouchers of widely used regional stores or companies (e.g. grocery stores), that would be likely to attract wide parts of the population and therefore ensure a representative sample. Participants were informed beforehand that the survey was completely anonymous and apart from demographic data, no personal data would be collected. For entering the raffle, par-ticipants were directed to a second survey, where the email-address was collected in-dependently from the main survey to ensure anonymity. For each country one voucher worth 100€ and five vouchers worth 20€ (or of equal worth in the corresponding na-tional currency) were raffled among those who gave their email-address.”
- "Prior to all analyses, respondents who were assumed to have answered items randomly were excluded to ensure data quality." - How was it verified that the respondents answered randomly?
Author response: Thank you for your comment. We followed the recommendations by Meade and Craig (2012) to identify careless respondents who have answered items randomly. The process is already described in detail in the paragraph following the cited sentence (l. 177): “Following Meade and Craig [49], several indicators were determined for each participant to identify these so-called careless respondents by using the ‘careless’ package in R4.0.2 [50], including the duration of participation, LongString and the Psychometric Synonyms Index as overall measures as well as intraperson variance and Mahalanobis distance for each separate scale. Participants, who were flagged on one or more indicators, were rated independently by two of the authors (CM, CKa). When careless data patterns became apparent, these persons were excluded from the study. In case of disagreement, a consensus was reached by discussion. “
- Lines 176: The authors should describe the abbreviation of SPSS 27.
Author response: Thank you for your comment. We have added the full name of the software and the reference.
Changes in manuscript:
Methods (l. 186): “To avoid listwise deletion, missing values were dealt with using expectation-maximization-based single value imputation, as implemented in the Statistical Package für Social Sciences (SPSS), Version 27.0 [52].“
- In Results section the authors should not start sentences with "N = ...".
Author response: Thank you for your comment. We have changed the wording as you have suggested.
Changes in manuscript:
Methods (l. 235): “The vignette was shown to N = 853 participants. Of those, 97 did not complete the survey and another 19 participants were excluded for giving careless responses.”
- The authors should describe the tables.
Author response: Thank you for this comment. In our opinion, all tables have been described in detail and we do not see the need for further descriptions. However, if you have specific suggestions, we are happy to add that information in the manuscript.
- Some conclusions should be transferred to limitations and implications to practice.
Author response: Thank you for this comment. We have transferred the idea of including more than two variables into the analyses from conclusion into the limitations section. We have also added some implications for practice in the conclusion.
Changes in manuscript:
Limitations: (l. 541): “Finally, for an in-depth understanding of possible interactions between country and gender, we chose a narrow focus for our analyses and the chosen method did not allow to include further cultural or sociodemographic covariates.”
Conclusions: (l. 558): “Also, practitioners should be aware of potential gender differences in causal beliefs and consider these and potential interactions with cultural factors in psychotherapy and counseling and especially when culturally adapting interventions.”
Reviewer 4 Report
Meyer et. al., report a single survey vignette-based study of the causal beliefs of PTSD in a cross-cultural sample from 5 countries. Authors here addressed a complicated, but important research question to understand the association between factors which influence cultural differences and shape the causal beliefs of lay person. Findings from such and more future studies could help illustrate ways of developing and providing culturally appropriate clinical services for different ethnic groups.
Authors provided strong justification for their hypotheses and study design. They conducted a thorough and appropriate analysis by combining both qualitative and quantitative approaches. One of their main findings was that participants held a wide variety of casual beliefs, often overlapping and of multifactorial nature. However, these factors resembled causes for PTSD that have been identified by other meta-analyses. Another interesting but important finding highlights the importance gender-aspects in multi-cultural research. They found significantly difference in beliefs between women and men and the differences between countries were more pronounced for women compared to men.
There are certainly few limitations, but authors have identified those and properly addressed in the discussion (e.g., balance concerning gender, age and socio-economic status of participants). It would have been interesting to see the effects on the second-order themes if male participants were presented with a female case vignette and vice versa. Furthermore, the limitation of the linguistic details in such multinational study can be minimized by a more collaborative approach whereby including researchers from the participating countries would help with translation and normalizing the biases. I think one another major limitation of this cross-cultural research was the choice of country where the commonalities seem to have confounded the results. Culturally, as these countries are not on extreme ends, it potentially explains the current finding of considerable overlap between the five countries. However, this choice certainly has higher application.
Authors have cited appropriate references through the manuscript.
Overall, the current manuscript is excellent and presented in a well-structured manner, and I think it makes a very needed contribution to the field. I do not have any revisions to suggest.
Author Response
Reviewer 4: Comments and Suggestions for Authors
Meyer et. al., report a single survey vignette-based study of the causal beliefs of PTSD in a cross-cultural sample from 5 countries. Authors here addressed a complicated, but important research question to understand the association between factors which influence cultural differences and shape the causal beliefs of lay person. Findings from such and more future studies could help illustrate ways of developing and providing culturally appropriate clinical services for different ethnic groups.
Authors provided strong justification for their hypotheses and study design. They conducted a thorough and appropriate analysis by combining both qualitative and quantitative approaches. One of their main findings was that participants held a wide variety of casual beliefs, often overlapping and of multifactorial nature. However, these factors resembled causes for PTSD that have been identified by other meta-analyses. Another interesting but important finding highlights the importance gender-aspects in multi-cultural research. They found significantly difference in beliefs between women and men and the differences between countries were more pronounced for women compared to men.
There are certainly few limitations, but authors have identified those and properly addressed in the discussion (e.g., balance concerning gender, age and socio-economic status of participants). It would have been interesting to see the effects on the second-order themes if male participants were presented with a female case vignette and vice versa. Furthermore, the limitation of the linguistic details in such multinational study can be minimized by a more collaborative approach whereby including researchers from the participating countries would help with translation and normalizing the biases. I think one another major limitation of this cross-cultural research was the choice of country where the commonalities seem to have confounded the results. Culturally, as these countries are not on extreme ends, it potentially explains the current finding of considerable overlap between the five countries. However, this choice certainly has higher application.
Authors have cited appropriate references through the manuscript.
Overall, the current manuscript is excellent and presented in a well-structured manner, and I think it makes a very needed contribution to the field. I do not have any revisions to suggest.
Author response: Thank you very much for your positive evaluation of our manuscript. We greatly appreciate the time and effort taken for this review.
Reviewer 5 Report
I recommend the manuscript entitled "Do gender and country of residence matter? A mixed methods 2 study on lay causal beliefs about PTSD" for publication in IJERPH. The paper concerns topical mental health issues. The findings it reports should be taken into account in both PTSD therapy and prophylaxis.
Author Response
Reviewer 5: Comments and Suggestions for Authors
I recommend the manuscript entitled "Do gender and country of residence matter? A mixed methods 2 study on lay causal beliefs about PTSD" for publication in IJERPH. The paper concerns topical mental health issues. The findings it reports should be taken into account in both PTSD therapy and prophylaxis.
Author response: Thank you very much for your positive evaluation of our manuscript. We greatly appreciate the time and effort taken for this review.
Round 2
Reviewer 2 Report
Congratulations for the work.
Reviewer 3 Report
Dear authors,
thank you for the corrections made.
The subject of the work is relevant to the special issue: Defining, Assessing, and Treating Stress-Related Disorders across Cultures. The manuscript is of good impact, well conceptualized, and clearly written.
Thank you